# Effects of Machining Induced Residual Shear and Normal Stresses on Fatigue Life and Stress Intensity Factor of Inconel 718

**Yang Hua** [1,2] **and Zhanqiang Liu** [1,2,*]

[1] School of Mechanical Engineering, Shandong University, Jinan 250061, China; sduhuayang@gmail.com
[2] Key Laboratory of High Efficiency and Clean Mechanical Manufacture of MOE/Key National Demonstration Center for Experimental Mechanical Engineering Education, Shandong University, Jinan 250061, China
[*] Correspondence: melius@sdu.edu.cn; Tel.: +86-531-8839-3206; Fax: +86-0531-8839-2045

**Abstract:** Residual shear stresses and normal stresses induced by machining affect the fatigue performance of components. Thus, residual shear and normal stresses should be considered simultaneously when evaluating the influence of residual stress on fatigue performance. In the present paper, the influences of residual shear and normal stresses on the fatigue life and stress intensity factor (SIF) of turned Inconel 718 were investigated. Firstly, the cos $\alpha$ measurement method was utilized to calculate the residual shear stress and residual normal stress of turned Inconel 718. Then, the combined effects of residual shear and normal stresses on fatigue life were evaluated through uniaxial tension–tension fatigue tests. Thirdly, a prediction model for the SIF was proposed by taking the residual shear and normal stresses into account. Finally, the predicted SIF was validated by the published experimental data from the literature. The predicted results of the proposed model generally agreed well with the available experimental data.

**Keywords:** residual normal stress; residual shear stress; fatigue; threshold stress intensity factor; Inconel 718

---

## 1. Introduction

Inconel 718 has good mechanical properties such as high yield and high ultimate tensile strength, fatigue resistance as well as good corrosion resistance at a high temperature of 650 °C. Thus, it has been widely employed in the aerospace industry for parts in turbine engines such as turbine disks, turbine shafts, and high-pressure compressor blades [1]. Turbine engines work with maximum speed in a harsh environment with high pressure, high load, and high temperature. Once a failure is caused by a fatigue fracture occurring on the turbine disk, turbine shaft, or compressor blade, a larger amount of high-energy debris will be generated. These debris will break through the turbine casing to threaten the aircraft safety and may cause catastrophic failure of the turbine.

The surface integrity factors including surface roughness, residual stress, micro-hardness, and microstructure are considered the most relevant factors that will result in a failure of the turbine disk or shaft according to statistics [2]. It is noteworthy that, among the surface integrity factors, residual stress plays a key role in affecting the fatigue performance of components [3–6]. Compressive residual stress (CRS) is beneficial to improving fatigue performance [7–9], whereas induced tensile residual stress is usually detrimental to the fatigue life of components [10,11].

Several researchers have undertaken to study the influence of residual stress on the fatigue performance of Inconel 718. Chen et al. [12] investigated the influence of the broaching process on the bending fatigue behavior of Inconel 718. They pointed out that the CRS in a longitudinal direction at the

surface induced by broaching could extend the bending fatigue life of Inconel 718. Kattoura et al. [13] analyzed the effect of CRS produced by ultrasonic nanocrystal surface modification on the fatigue behavior of Inconel 718 plus superalloy. The experimental results showed that the presence of CRS in longitudinal direction and transverse direction improved fatigue life. They explained that the CRS at the surface layer could inhibit the crack initiation and propagation. Fleury et al. [14] evaluated the influence of residual stress in a longitudinal direction on the fatigue life of machined nickel-based superalloy. They found that higher fatigue life could be obtained due to the presence of CRS. The negative effect of notch on fatigue life could be canceled by the positive effect of CRS. Martin-Meizoso et al. [15] carried out machining and fatigue experiments to correlate residual stress with the fatigue life of Inconel 718 at high temperature. They suggested that the maximum residual stress at the machined surface and the area of CRS were related to fatigue life. Similar work on titanium alloy Ti6Al4V was performed by Moussaoui et al. [16,17]; they assumed that the maximum residual stress in a longitudinal direction on the machined surface could be a good indicator for evaluating fatigue life.

There is extensive literature focusing on the residual stresses in longitudinal or transverse directions on a machined surface. However, the residual stresses in these two directions are residual normal stresses, and the residual shear stress produced during machining has not been taken into account. Indeed, both the residual normal and shear stresses were induced during the machining process. The residual shear stress induced by machining should be taken into account to obtain the equivalent stress that can determine the fatigue life of a component. Hence, the induced residual shear stress is not negligible, and it is necessary to study its effect on fatigue performance of Inconel 718. In addition, the effect of residual stress on the stress intensity factor (SIF) has not been entirely explored and reported in the literature. Therefore, the purpose of this paper was to determine the residual shear and normal stresses simultaneously and to establish the correlation between the residual stress and the SIF of Inconel 718. The residual normal stress was obtained using X-ray diffraction and the residual shear stress was calculated based on the cos$\alpha$ method. Uniaxial tension–tension fatigue tests were carried out to evaluate their effect on fatigue life. Moreover, a prediction model of the stress intensity factor was proposed by taking the residual shear and normal stresses into account. The proposed model is validated by the published experimental data from the literature.

## 2. Theoretical Analysis

### 2.1. Residual Shear and Normal and Stresses Calculation

Figure 1a depicts the schematic diagram of non-uniform plastic deformation generation during the turning process. It is observed from Figure 1a that compressive, tensile, and shear plastic deformations were produced on the workpiece material ahead of the cutting tool during the turning process. On the other hand, a larger amount of heat was produced during machining which would lead to the compressive plastic deformation on the machined surface [1,5]. The interaction of these non-uniform plastic deformations determined the final state of residual stress as shown in Figure 1b. It should be noted that the *x* and *y* axes in Figure 1b correspond to the longitudinal and transverse directions, respectively.

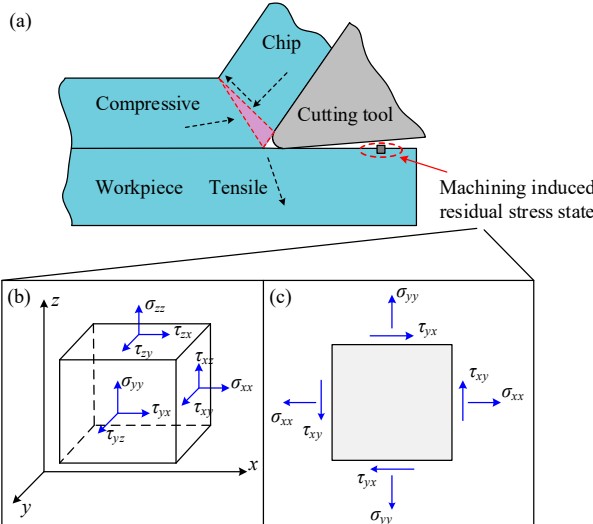

**Figure 1.** (**a**) Residual stress generation during machining, (**b**) space stress state, and (**c**) plane stress state.

The traditional X-ray diffraction based on the $\sin^2 \psi$ method was used to determine the residual stress according to Hooke's law of an isotropic solid material [18]:

$$\varepsilon_{\varphi\psi} = \frac{1+\nu}{E}\sigma_\varphi \sin^2 \psi - \frac{\nu}{E}\left(\sigma_{xx} + \sigma_{yy}\right) \tag{1}$$

$$\sigma_\varphi = \frac{E}{1+\nu}\frac{\partial \varepsilon_{\varphi\psi}}{\partial \sin^2 \psi} \tag{2}$$

where $\varepsilon_{\varphi\psi}$ is the strain at $(\varphi, \psi)$ direction, $E$ and $\nu$ are the Young's modulus and Poisson's ratio of material, and $\sigma_{xx}$ and $\sigma_{yy}$ are the stress components (normal stress). $\sigma_\varphi$ is the normal stress in the $\varphi$ direction.

In the above two equations, it is noteworthy that only the residual normal stress $\sigma_\varphi$ can be determined by using the $\sin^2 \psi$ method. In order to obtain the residual shear stress, the $\cos\alpha$ measurement technology proposed by Taria et al. [19] and modified by Sasaki et al. [20] was utilized. The relationship between the strain $\varepsilon_\alpha$ and the stress components in $x−y−z$ coordinates can be expressed by Equation (3):

$$\varepsilon_\alpha = \frac{1+\nu}{E}\left(\sigma_x n_1^2 + \sigma_y n_2^2 + \sigma_z n_3^2 + 2\tau_{xy}n_1 n_2 + 2\tau_{yz}n_2 n_3 + 2\tau_{zx}n_3 n_1\right) - \frac{\nu}{E}\left(\sigma_x + \sigma_y + \sigma_z\right) \tag{3}$$

where $\tau_{xy}$, $\tau_{yz}$, and $\tau_{zx}$ represent the shear stresses. $n_1$, $n_2$, and $n_3$ are the components of diffraction vector $n$ as shown in Figure 2 and can be obtained by Equation (4) [20]:

$$\begin{aligned}
n_1 &= \cos\eta\sin\psi\cos\varphi - \sin\eta\cos\psi\cos\varphi\cos\alpha - \sin\eta\sin\psi\sin\alpha \\
n_2 &= \cos\eta\sin\psi\sin\varphi - \sin\eta\cos\psi\sin\varphi\cos\alpha + \sin\eta\cos\psi\sin\alpha \\
n_3 &= \cos\eta\cos\psi + \sin\eta\sin\psi\cos\alpha
\end{aligned} \tag{4}$$

where $\alpha$ is a central angle that can vary from 0 to 90° in order to cover the whole ring, $\psi$ is the tilt angle between the incident beam and axial $z$, $2\eta$ is a complementary angle of diffraction angle, and $\varphi$ is the rotation angle in the $x−y$ plane as shown in Figure 2.

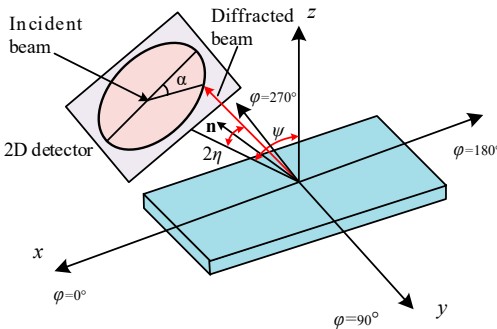

**Figure 2.** Residual stress determination by using cosα method.

According to the Equation (3), a set of four strains $\varepsilon_\alpha$, $\varepsilon_{\pi+\alpha}$, $\varepsilon_{-\alpha}$, and $\varepsilon_{\pi-\alpha}$ can be obtained which are used to calculate the following two strains: $\varepsilon_{\alpha1}$ and $\varepsilon_{\alpha2}$.

$$\varepsilon_{\alpha1} = [(\varepsilon_\alpha - \varepsilon_{\pi+\alpha}) + (\varepsilon_{-\alpha} - \varepsilon_{\pi-\alpha})]/2 \tag{5}$$

$$\varepsilon_{\alpha2} = [(\varepsilon_\alpha - \varepsilon_{\pi+\alpha}) - (\varepsilon_{-\alpha} - \varepsilon_{\pi-\alpha})]/2 \tag{6}$$

The X-ray diffraction measurements should be carried out at four rotation angles $\varphi$ (0, 90, 180, and 270° as illustrated in Figure 2) to determine the normal stress and shear stress.

For the case of $\varphi = 0°$, by substituting Equation (3) into Equations (5) and (6), the following equations can be obtained:

$$\varepsilon_{\alpha1,\varphi=0} = -\frac{1+\nu}{E}\left[(\sigma_{xx} - \sigma_{zz})\sin2\psi + \tau_{zx}\cos2\psi\right]\sin2\eta\cos\alpha \tag{7}$$

$$\varepsilon_{\alpha2,\varphi=0} = 2\frac{1+\nu}{E}\left[\tau_{xy}\sin\psi + \tau_{yz}\cos\psi\right]\sin2\eta\sin\alpha \tag{8}$$

From Equation (7), a linear relationship between $\varepsilon_{\alpha1}$ and cosα can be observed. Thus, the normal stress $\sigma_{\varphi=0}$ can be determined:

$$\sigma_{\varphi=0} = (\sigma_x - \sigma_z) + 2\tau_{zx}\cot2\psi = -\frac{E}{1+\nu}\frac{1}{\sin2\eta\sin2\psi}\frac{\partial\varepsilon_{\alpha1,\varphi=0}}{\partial\cos\alpha} \tag{9}$$

From Equation (8), a linear relationship between $\varepsilon_{\alpha1}$ and sinα can be observed. The shear stress $\tau_{\varphi=0}$ can be determined:

$$\tau_{\varphi=0} = \tau_{xy} + \tau_{yz}\cot\psi = \frac{E}{2(1+\nu)}\frac{1}{\sin2\eta\sin\psi}\frac{\partial\varepsilon_{\alpha2,\varphi=0}}{\partial\sin\alpha} \tag{10}$$

Similarly, the normal stress and shear stress can be determined for the case of $\varphi = 90°$, $\varphi = 180°$, and $\varphi = 270°$, respectively.

For the case of $\varphi = 90°$:

$$\sigma_{\varphi=90} = \left(\sigma_{yy} - \sigma_{zz}\right) + 2\tau_{yz}\cot2\psi = -\frac{E}{1+\nu}\frac{1}{\sin2\eta\sin2\psi}\frac{\partial\varepsilon_{\alpha1,\varphi=90}}{\partial\cos\alpha} \tag{11}$$

$$\tau_{\varphi=90} = -\tau_{xy} - \tau_{zx}\cot\psi = \frac{E}{2(1+\nu)}\frac{1}{\sin2\eta\sin\psi}\frac{\partial\varepsilon_{\alpha2,\varphi=90}}{\partial\sin\alpha} \tag{12}$$

For the case of $\varphi = 180°$:

$$\sigma_{\varphi=180} = (\sigma_{xx} - \sigma_{zz}) - 2\tau_{zx}\cot 2\psi = -\frac{E}{1+v}\frac{1}{\sin 2\eta \sin 2\psi}\frac{\partial \varepsilon_{\alpha 1,\varphi=180}}{\partial \cos \alpha} \tag{13}$$

$$\tau_{\varphi=180} = -\tau_{xy} - \tau_{zx}\cot \psi = \frac{E}{2(1+v)}\frac{1}{\sin 2\eta \sin \psi}\frac{\partial \varepsilon_{\alpha 2,\varphi=180}}{\partial \sin \alpha} \tag{14}$$

and for the case of $\varphi = 270°$:

$$\sigma_{\varphi=270} = \left(\sigma_{yy} - \sigma_{zz}\right) - 2\tau_{yz}\cot 2\psi = -\frac{E}{1+v}\frac{1}{\sin 2\eta \sin 2\psi}\frac{\partial \varepsilon_{\alpha 1,\varphi=270}}{\partial \cos \alpha} \tag{15}$$

$$\tau_{\varphi=270} = -\tau_{xy} + \tau_{zx}\cot \psi = \frac{E}{2(1+v)}\frac{1}{\sin 2\eta \sin \psi}\frac{\partial \varepsilon_{\alpha 2,\varphi=270}}{\partial \sin \alpha} \tag{16}$$

Consequently, stress components can be determined by the following equations:

$$\sigma_{xx} - \sigma_{zz} = \left[\sigma_{\varphi=0} + \sigma_{\varphi=180}\right]/2 \tag{17}$$

$$\sigma_{yy} - \sigma_{zz} = \left[\sigma_{\varphi=90} + \sigma_{\varphi=270}\right]/2 \tag{18}$$

$$\tau_{xy} = \left[\tau_{\varphi=0} + \tau_{\varphi=180}\right]/2 \tag{19}$$

As the penetration depth of X-rays for Inconel 718 is about 5 μm during measurements [1,5], the stress state is usually assumed to be a plane stress state as illustrated in Figure 1c. The normal stress in the *z*-axis is zero. Therefore, the normal stress and shear stress can be determined by Equations (17)–(19).

## 2.2. Modeling for Stress Intensity Factor

In the existing literature, Moussaoui et al. [16,17] suggested that the maximum local stress $\sigma_{local,max}$ at the machined surface was considered a good indicator for evaluating fatigue life. They defined the stress $\sigma_{local,max}$ as the following Equation (20):

$$\sigma_{local,max} = \sigma_{applied,max} + \sigma_{RS,surface} \tag{20}$$

where $\sigma_{applied,max}$ represents the maximum stress that is applied on a specimen during fatigue tests. $\sigma_{RS,surface}$ is the residual stress measured in the axial direction at the machined surface. They proposed a new law that described the relationship between the maximum local stress $\sigma_{applied,max}$ and fatigue life $N_f$:

$$\sigma_{local,max} = 1.535 \times 10^8 \times N_f^{-1.105} + 362.2 \tag{21}$$

However, the residual shear stress is not taken into account to evaluate the effect of residual stress on fatigue life in Equations (20) and (21). As analyzed in Section 1, residual shear stress is produced during machining, and it should be considered when determining the equivalent stress. When the stress $\sigma_{applied}$ is applied in the axial direction of a machined specimen during fatigue tests, the equivalent stress can be calculated following the von Mises equation:

$$\sigma_v = \frac{1}{\sqrt{2}}\sqrt{(\sigma_1 - \sigma_2)^2 + (\sigma_2 - \sigma_3)^2 + (\sigma_3 - \sigma_1)^2} \tag{22}$$

where $\sigma_v$ is the equivalent stress. $\sigma_1$, $\sigma_2$, and $\sigma_3$ are the principal stresses which can be determined by Equation (23) for a plain stress condition:

$$\sigma_{1,2} = \frac{(\sigma_{xx}+\sigma_{applied})+\sigma_{yy}}{2} \pm \sqrt{\left(\frac{(\sigma_{xx}+\sigma_{applied})-\sigma_{yy}}{2}\right)^2 + \tau_{xy}^2}$$
$$\sigma_3 = 0$$
(23)

Once the equivalent stress is calculated by the Equation (22), the stress intensity factor (SIF) can be determined according to the facture of the specimen. Figure 3 exhibits the schematic diagram of the fracture surfaces for surface failure. The region with small facets is generally regarded as a rough area where the crack is prone to initiate. Hence, the region with small facets is defined as the initiation area and the crack shape is assumed to be semi-elliptical. For the crack initiation area, the value of the SIF range, $\Delta K_{ini}$, determines whether the crack propagates or not. Consequently, the value of $\Delta K_{ini}$ can be considered as the threshold for crack propagation $\Delta K_{th}$ [21]. For a cylinder specimen, the value of the SIF range $\Delta K_{ini}$ under tension–tension fatigue test can be given [21]:

$$\Delta K_{ini} = \Delta K_{th} = n \cdot \Delta\sigma \cdot \sqrt{\pi \cdot \sqrt{area}}$$
(24)

where $n$ is the correction factor which is related to the geometry of the crack. For a crack initiated at the surface, $n = 0.65$ [21]. $\sqrt{area}$ is the equivalent size of the projected area which includes facets. $\Delta\sigma$ is the applied stress range and can be expressed by Equation (25):

$$\Delta\sigma = \sigma_{applied,\max} - \sigma_{applied,\min}$$
(25)

Due to the presence of residual stress, the effective stress $\Delta\sigma_{eff}$ can be obtained according to Equation (19) [16,17].

$$\Delta\sigma_{eff} = \left(\sigma_{applied,\max} + \sigma_{RS,surface}\right) - \left(\sigma_{applied,\min} + \sigma_{RS,surface}\right) = \Delta\sigma$$
(26)

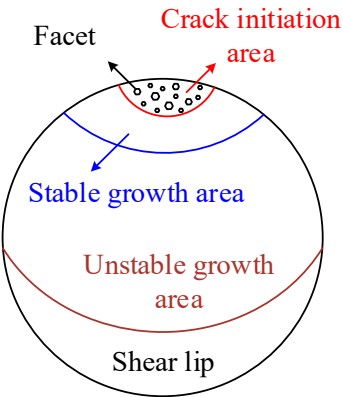

**Figure 3.** Schematic diagram of surfaces failure.

Based on the description of Equations (25) and (26), it is interesting to find that Equation (26) is equal to Equation (25). It is indicated that the presence of residual stress has no influence on the effective stress that acted on the specimen. It can be deduced that fatigue life will not be changed when residual stress is introduced by machining. Thus, Equation (20), proposed by Moussaoui et al. [16,17], may not be suitable to describe the effect of residual stress on the SIF range. Consequently, in this paper, a new effective stress range $\Delta\sigma'_{eff}$ is introduced by Equation (27):

$$\Delta\sigma'_{eff} = \sigma_{v,\text{max}} - \sigma_{v,\text{min}} \tag{27}$$

Thus, Equation (24) can be rewritten as:

$$\Delta K_{th} = n \cdot \Delta\sigma'_{eff} \cdot \sqrt{\pi \cdot \sqrt{area}} = n \cdot (\sigma_{v,\text{max}} - \sigma_{v,\text{min}}) \cdot \sqrt{\pi \cdot \sqrt{area}} \tag{28}$$

where the maximum and minimum equivalent stress can be calculated by Equations (29) and (30).

$$\sigma_{v,\text{max}} = \sqrt{\left(\sigma_{xx} + \sigma_{applied,\text{max}} - \sigma_{yy}\right)^2 + \left(\sigma_{xx} + \sigma_{applied,\text{max}}\right)\sigma_{yy} + 3\tau_{xy}^2} \tag{29}$$

$$\sigma_{v,\text{min}} = \sqrt{\left(\sigma_{xx} + \sigma_{applied,\text{min}} - \sigma_{yy}\right)^2 + \left(\sigma_{xx} + \sigma_{applied,\text{min}}\right)\sigma_{yy} + 3\tau_{xy}^2} \tag{30}$$

## 3. Materials and Methods

### 3.1. Workpiece Material and Specimen Preparation

Inconel 718 was employed as the workpiece material in this paper. The as-received material was a wrought cylinder bar with a diameter of 130 mm and a length of 300 mm. The material was solution heat treated and age treated. The recommended solution heat treatment was performed on a furnace at 960 °C for 1 h and then air cooled to room temperature. After the solution heat treatment, the age treatment was conducted on a furnace at 720 °C for 8 h and then furnace cooled to 620 °C with a cooling rate of 50 °C/h, held at 620 °C for 8 h, and finally air cooled to room temperature. The material was divided into $\Phi$14 mm $\times$ 140 mm cylinder bars using wire cutting electrical discharge machining (WEDM). The chemical compositions of the material are summarized in Table 1 [22]. The mechanical properties of the material at room temperature are shown in Table 2 [22].

**Table 1.** Chemical composition of the material (% wt) [22].

| Cr | Nb | Al | Ni | Mn | Mo | Cu | Ti | Co | Si |
|------|------|------|------|------|------|------|------|------|------|
| 18.05 | 5.43 | 0.50 | 53.51 | 0.062 | 2.98 | 0.035 | 1.02 | 0.31 | 0.074 |
| Sn | Ta | N | C | P | B | Ca | Mg | S | Fe |
| 0.00091 | 0.0085 | 0.0079 | 0.025 | 0.010 | 0.0042 | 0.0032 | 0.0014 | 0.00078 | Bal |

**Table 2.** Tensile mechanical properties of the material at room temperature [22].

| Elastic Modulus E (GPa) | Ultimate Tensile Strength $\sigma_b$ (MPa) | Yield Strength $\sigma_{0.2}$ (MPa) | Elongation (%) | Hardness (HBW) |
|------|------|------|------|------|
| 205 | 1502 | 1360.5 | 19.3 | 439 |

Four different turning processes were carried out to evaluate the influence of residual shear and normal stress on fatigue life and SIF. The turning tool VBMT 110308-1105 with a tool nose radius of 0.8 mm was employed. All experiments were performed on a CNC machining center with a fixed feed per revolution at $f$ = 0.075 mm/rev and a depth of cut at $a_p$ = 0.2 mm under various cutting speeds ($Vc$) as summarized in Table 3. The geometry of the fatigue specimens in the present study is shown in Figure 4.

**Table 3.** Turning conditions.

| Condition | Cutting Speed $Vc$ (m/min) | Feed Rate $f$ (mm/rev) | Depth of Cut $a_p$ (mm) | Tool Nose Radius $r$ (mm) |
|---|---|---|---|---|
| SI | 50 | | | |
| SII | 60 | 0.075 | 0.2 | 0.8 |
| SIII | 70 | | | |
| SIV | 80 | | | |

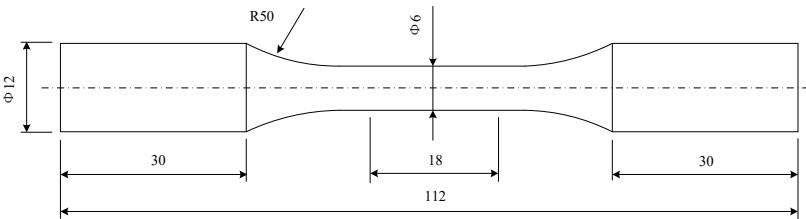

**Figure 4.** The geometry of fatigue test specimen (unit: mm).

### 3.2. Residual Stress Measurements

The X-ray diffraction equipment Pulstec μ-X360n was used to measure residual stress based on the cos$\alpha$ method. All measurements were conducted on the crystallographic plane {311} using the Kβ wavelength. All measurements were conducted with the same parameters, which are summarized in Table 4, in order to guarantee the accuracy of the measurements. As mentioned in Section 2, residual stress in four rotation angles ($\varphi$ = 0, 90, 180, and 270°) should be measured to determine the shear stress and normal stress.

**Table 4.** X-ray diffraction parameters of residual stress measurements.

| Specimen | Tube target | Crystallographic plane | Diffraction angle (2θ) |
|---|---|---|---|
| Inconel 718 | Cr | {311} | 150.89° |
| **Tube voltage** | **Tube current** | **X-ray slit** | **Exposure time** |
| 30 kV | 1 mA | 2 mm | 90 s |

### 3.3. Fatigue Tests

Uniaxial tension–tension fatigue tests were carried out to study the influence of residual stress on the fatigue performance of Inconel 718. All the turned specimens were polished by hand to reduce the influence of surface roughness (Ra) on fatigue life even though the experimental results in the literature [16,17,22] concluded that, compared with residual stress, the surface roughness had a less significant role on fatigue life. All fatigue tests were performed on a fatigue testing machine PLG-100 with a maximum frequency of 250 Hz at room temperature. A dynamic sinusoidal load was applied on the fatigue specimen with a maximum stress of 1131 MPa during the fatigue tests. The frequency of the fatigue test was approximately 110 Hz, and the stress ratio was $R = \sigma_{min}/\sigma_{max} = 0.1$. Six fatigue specimens were tested for each cutting condition to eliminate random error. The fracture surface of the fatigue specimen was analyzed using scanning electron microscopy (SEM).

## 4. Results and Discussion

### 4.1. Residual Shear and Normal Stress Analysis

The residual stresses at the surface in four rotation angles (i.e., $\sigma_{\varphi = 0}$, $\sigma_{\varphi = 90}$, $\sigma_{\varphi = 180}$, and $\sigma_{\varphi = 270}$) were measured using an X-ray diffractometer for each machined specimen. The surface residual

normal stresses $\sigma_{xx}$ and $\sigma_{yy}$, calculated by Equations (17) and (18), and the residual shear stress, $\tau_{xy}$, calculated by Equation (19), are summarized in Figure 5.

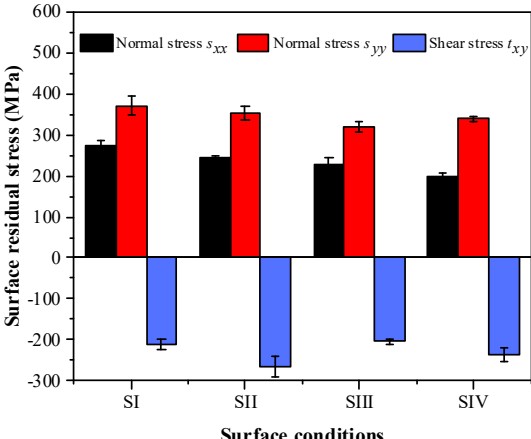

**Figure 5.** Residual normal and shear stresses calculated using the cos$\alpha$ method.

From Figure 5, it is observed that the residual normal stresses $\sigma_{xx}$ were tensile and the values varied within the range of 199~275 MPa, the residual normal stresses $\sigma_{yy}$ were tensile and the values within ranges of 320~370 MPa, and the residual shear stresses $\tau_{xy}$ were negative with the value changes in a range of −265~−211.5. The value of residual shear stress calculated by Equation (19) supported the previous statement that the machining process induced the residual shear stress. A similar phenomenon was found by Hananbusa et al. [23]. Kamura et al. [24] reported that residual shear stress had been observed on a bearing surface fatigued by rolling contact.

It is noted that the residual normal stresses $\sigma_{xx}$ and $\sigma_{yy}$ decreased slightly as the cutting speed increased. This drop in residual normal stresses was mainly due to the increased chip flow rate. The increased cutting speed resulted in a higher chip flow rate and reduced the contact time between the tool and the workpiece material. Consequently, the amount of thermal energy was taken away by the chip increase [25]. It is supposed that the heat generated in the primary shear zone, tool-chip contact zone, and tool-workpiece friction zone diffused into the workpiece surface decreased. Therefore, the residual normal stress decreased with the increase in cutting speed. The residual shear stresses $\tau_{xy}$ had no obvious and orderly variation as the cutting speed increased. A similar phenomenon was found by Devillez et al. [26] who concluded that the maximum value of tensile residual stress in longitudinal and transverse directions decreased when the cutting speed increased under dry cutting conditions. They explained that this phenomenon was attributed to the reduction of heat generated due to the increased cutting speed. Nevertheless, Moussa et al. [27] point out that tensile residual stress increased significantly in longitudinal and transverse directions as the cutting speed increased under a low cutting speed ($V_c < 100$ m/min). They explained that the low cutting speed ($V_c < 100$ m/min) would lead to a low material removal rate (MRR). This resulted in a decrease in the thermal dissipation by the chip. Therefore, more heat generation would be transmitted to the surface of the workpiece resulting in a higher tensile residual stress at the surface.

### 4.2. Fatigue Life Analysis

Once the residual shear stress and residual normal stress were determined, fatigue tests were performed to explore the effect on fatigue life. Under the same fatigue testing condition, the fatigue life of a specimen with different levels of residual shear stress and residual normal stress are presented in Figure 6a–c. The fatigue test results illustrated that no evident fatigue life evolution tendencies were observed with the increase in the residual normal and shear stresses. It is suggested that the fatigue life of a specimen was not directly dependent upon only residual shear stress or residual normal stresses.

Consequently, it can be deduced that residual stress in only one direction would not be a suitable indicator for evaluating the effect of residual stress on fatigue life.

Figure 6d illustrates the relationship between the equivalent stress and fatigue life. Compared with the residual shear stress and residual normal stress, the equivalent stress significantly influenced the fatigue life of the specimen. It is clear from Figure 6d that there was a linear relationship between the equivalent stress and fatigue life. The longest and shortest fatigue lives (mean value) of the machined specimens were 59,765 cycles and 36,236 cycles which correspond to the equivalent stress at the levels of 1248 MPa and 1317 MPa, respectively. As the equivalent stress decreased from 1317 to 1248 MPa, the fatigue life tended to increase from 36,236 cycles to 59,765 cycles, a 39.4% increase (see Figure 6d).

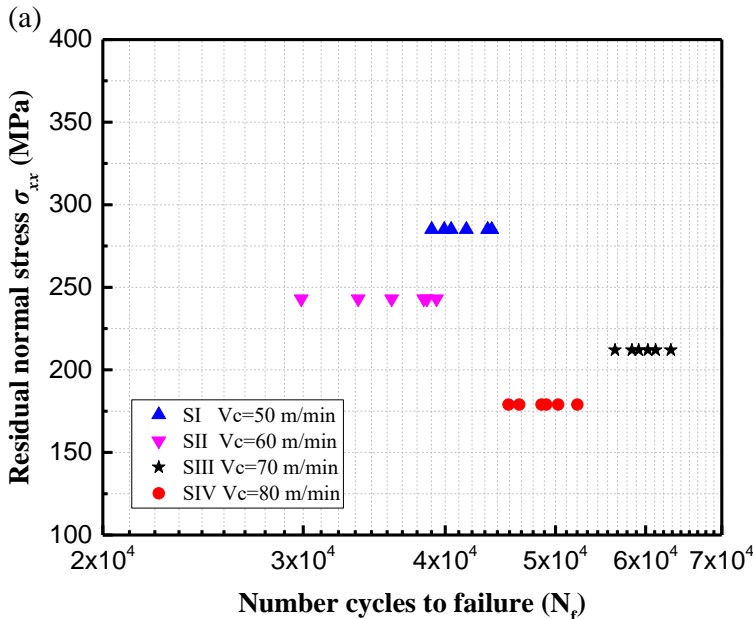

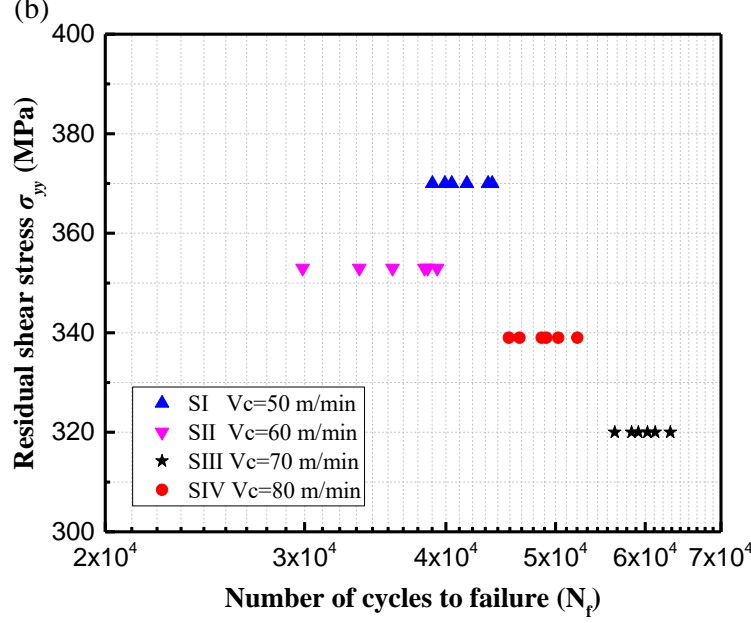

**Figure 6.** *Cont.*

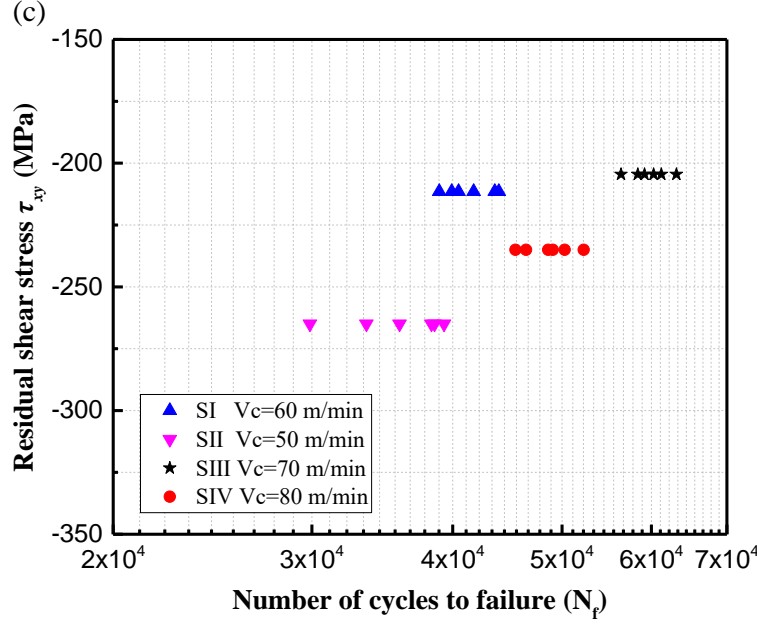

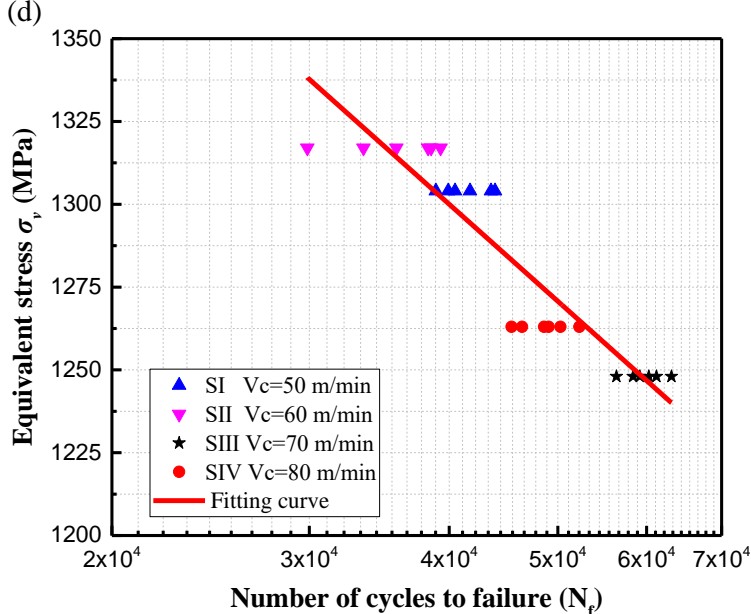

**Figure 6.** Fatigue life at different stresses: (**a**) residual normal stress $\sigma_{xx}$, (**b**) residual normal stress $\sigma_{yy}$, (**c**) residual shear stress $\tau_{xy}$, and (**d**) equivalent stress $\sigma_v$.

To further understand the effect of residual stress on fatigue life, a comparison was made between the equivalent stress calculated by Equation (22) and the total stresses calculated by Equation (20) according to Moussaoui et al. [16,17]. The relationship between the calculated total stresses and the fatigue life were plotted in Figure 7. It is observed that the results of the total stresses were depicted as a function of fatigue life. The maximum total stress calculated by Equation (20) was nearly 1400 MPa, which exceeded the yield strength ($\sigma_{0.2} = 1360.5$ MPa) of the material. However, the maximum total stress calculated by the proposed model was less than the yield strength, which corresponded to a fatigue life of $3.6 \times 10^4$ cycles. From Figure 7, it is noted that the higher total stress calculated by Equation (22) gave rise to the lower fatigue life of the machined specimen. However, as the total stress calculated by Equation (20) increased, there was no evident tendency of the fatigue life to diminish gradually. This phenomenon was not consistent with Moussaoui et al. [16,17] who reported

that the maximum value of residual stress at the surface was a good indicator for evaluating fatigue performance as summarized in Figure 8. In the figure, it can be found that the fatigue life increased with the decrease in the maximum local stress. Nevertheless, it was found in detail that the longest fatigue life did not occur at the largest compressive residual stress. (e.g., the fatigue life induced by the compressive stress of Specimen 4 was much shorter than that induced by Specimen 1, and Specimen 17 generated a longer fatigue life than Specimen 3). Thus, this indicated that the residual stress at the surface in only one direction might not be enough to accurately evaluate fatigue performance. In the present work, the fatigue tests revealed that there was no clear and orderly tendency of the residual stress at the surface to directly impact on fatigue life. The residual stress at the surface in one direction could not fully describe the effect of residual stress on fatigue life. Therefore, it can be inferred that the equivalent stress that incorporated residual shear and normal stresses might be more suitable for describing the fatigue life of machined components.

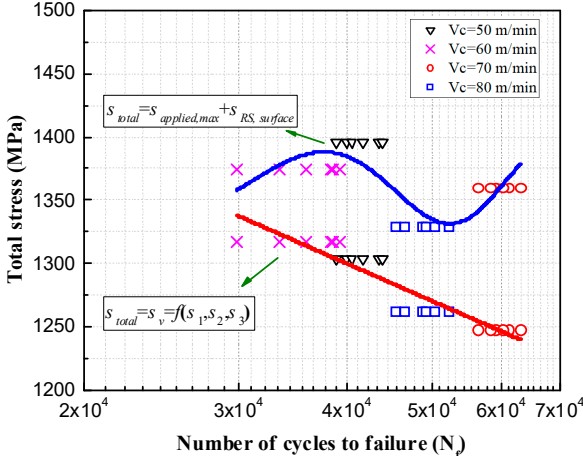

**Figure 7.** The relationship between the fatigue life and the total stress.

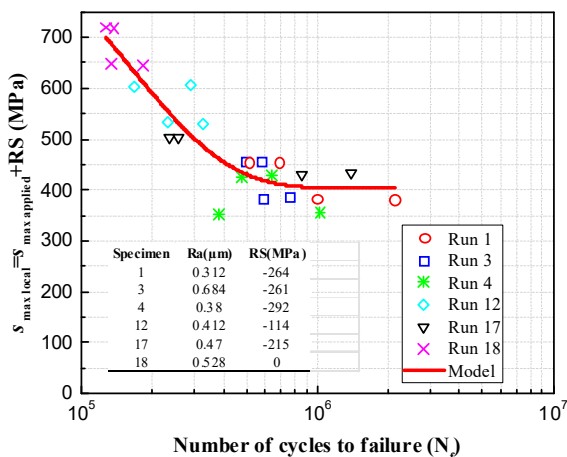

**Figure 8.** Residual stresses versus fatigue life: $\sigma_{\max\ local} = \sigma_{\max\ applied} + RS$.

### 4.3. Evaluation of SIF

The fatigue fracture morphologies of the specimens under different machining surface conditions were observed using SEM as presented in Figure 9. From the figure, it was found that all specimens exhibited a similar fracture surface pattern. The crack initiation occurred from the surface of the specimens and then propagated into the interior of the machined surface. A similar phenomenon was observed by Klotz et al. [28] who found that the crack initiated from the surface of machined Inconel 718 and presented multiple crack sources. Figure 10a shows a typical crack initiation region

under higher magnification of the SI surface condition. Figure 10b exhibits small facets in the crack initiation region. The facet area, the maximum equivalent stress $\sigma_{v,\max}$, the minimum equivalent stress $\sigma_{v,\min}$, and the effective stress range $\Delta\sigma'_{eff}$ for each machining condition are summarized in Table 5. The threshold stress intensity factor $\Delta K_{th}$ was obtained according to Equation (28).

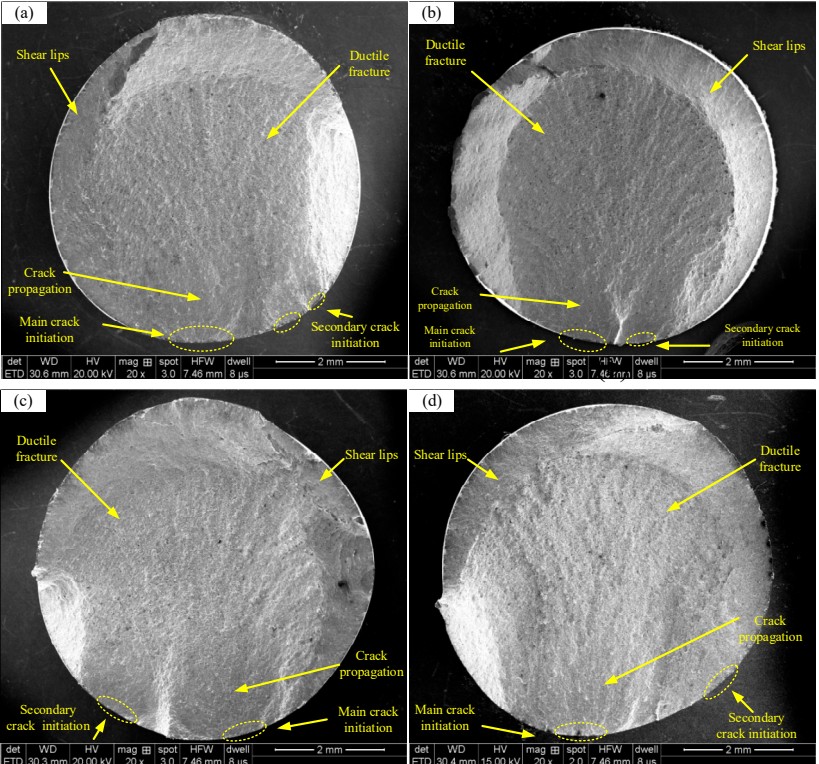

**Figure 9.** Fracture morphology of the specimen under different surface conditions: (**a**) SI, (**b**) SII, (**c**) SIII, and (**d**) SIV.

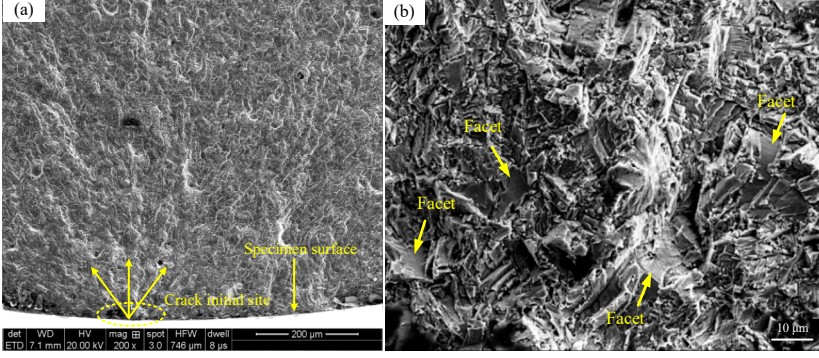

**Figure 10.** Crack initiation region under the SI surface condition: (**a**) crack initiation site and (**b**) small facets.

**Table 5.** The facet area of specimen fracture surfaces under different machining conditions.

| Machining Conditions | No. | Facet Area (mm²) | Moussaoui's Model [16,17] | | | The Proposed Model | | |
|---|---|---|---|---|---|---|---|---|
| | | | $\sigma_{local,max}$ | $\sigma_{local,min}$ | $\Delta\sigma_{eff}$ | $\sigma_{v,max}$ | $\sigma_{v,min}$ | $\Delta\sigma'_{eff}$ |
| $Vc = 50$ m/min | 1 | 0.05854 | 1406 | 388.1 | 1017.9 | 1304 | 523.6 | 780.4 |
| | 3 | 0.05526 | 1406 | 388.1 | 1017.9 | 1304 | 523.6 | 780.4 |
| | 4 | 0.05148 | 1406 | 388.1 | 1017.9 | 1304 | 523.6 | 780.4 |
| $Vc = 60$ m/min | 7 | 0.03014 | 1374 | 356.1 | 1017.9 | 1317 | 579.9 | 737.1 |
| | 9 | 0.04876 | 1374 | 356.1 | 1017.9 | 1317 | 579.9 | 737.1 |
| | 12 | 0.04198 | 1374 | 356.1 | 1017.9 | 1317 | 579.9 | 737.1 |
| $Vc = 70$ m/min | 13 | 0.07619 | 1374 | 342.1 | 1017.9 | 1248 | 485 | 763 |
| | 14 | 0.07061 | 1360 | 342.1 | 1017.9 | 1248 | 485 | 763 |
| | 17 | 0.08001 | 1360 | 342.1 | 1017.9 | 1248 | 485 | 763 |
| $Vc = 80$ m/min | 19 | 0.06538 | 1330 | 312.1 | 1017.9 | 1264 | 521 | 743 |
| | 21 | 0.06297 | 1330 | 312.1 | 1017.9 | 1264 | 521 | 743 |
| | 24 | 0.05689 | 1330 | 312.1 | 1017.9 | 1264 | 521 | 743 |

The values of $\Delta K_{th}$ for the Moussaoui model and the proposed model can be calculated according to Equations (26) and (28), respectively. Figure 11 shows the relationship between the fatigue life $N_f$ and SIF range $\Delta K_{th}$. From the figure, it can be deduced that the $\Delta K_{th}$ calculated by both Moussaoui et al. [16,17] and the proposed model maintained a constant regardless of fatigue life. It can be seen that the values of $\Delta K_{th}$, calculated by the Moussaoui's model, were in the range of 15.45~19.72 MPa $\sqrt{m}$ regardless of fatigue life. The mean value of $\Delta K_{th}$ was 18.11 MPa $\sqrt{m}$ approximately. Whereas the values of $\Delta K_{th}$, calculated by the proposed model, were in the range of 11.19~14.78 MPa $\sqrt{m}$ regardless of fatigue life. The mean value of $\Delta K_{th}$ was approximately 13.45 MPa $\sqrt{m}$, which is much lower than the calculated result by Moussaoui's model.

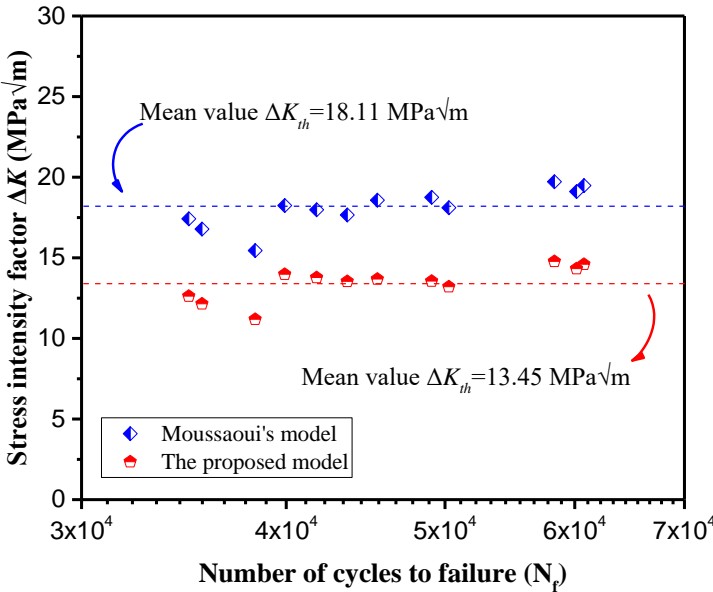

**Figure 11.** The relationship between the threshold stress intensity factor $\Delta K_{th}$ and the fatigue life.

Figure 12 shows the comparison of the crack propagation curves of Inconel 718 at the same stress ratio $R = 0.1$. Mercer et al. [29] revealed that the threshold value of the SIF range was approximately

13.25 MPa $\sqrt{m}$ which coincides with the threshold value in the range of 13.21~13.89 MPa $\sqrt{m}$ obtained by Clavel et al. [30]. Nevertheless, the value of $\Delta K_{th}$ obtained by Osinkolu et al. [31] was approximately 21.39 MPa $\sqrt{m}$ which is higher than that obtained by Mercer [29] and Clavel [30]. This could be explained by the fact that the experiments were performed in vacuum and under the lower stress ratio $R = 0.05$. Yamada and Newman [32,33] reported a threshold value $\Delta K_{th}$ that was approximately within the range of 11~12 MPa $\sqrt{m}$ at the stress ration $R = 0.1$ for Inconel 718. It can be noted that the yield stress $\sigma_{0.2}$ of Inconel 718 employed in References [32,33] was 1060 MPa which is much lower than the Inconel 718 used in this paper ($\sigma_{0.2} = 1360.5$ MPa). Thus, the threshold value $\Delta K_{th}$ obtained by Reference [32,33] is a little conservative. In the present work, the mean value of $\Delta K_{th}$ (13.45 MPa $\sqrt{m}$), calculated by the proposed model agreed well with the threshold value that was demonstrated by Mercer [29] and Clavel [30].

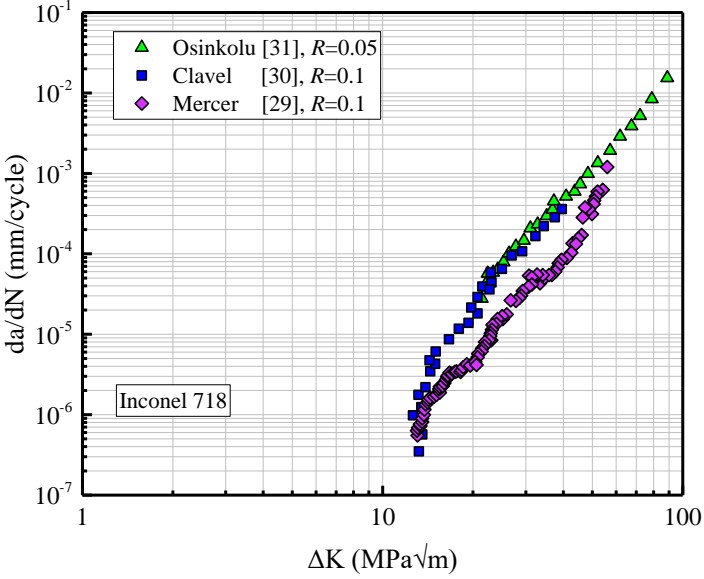

**Figure 12.** Fatigue crack growth rate at a stress ratio of 0.1 for Inconel 718.

The calculated results showed that the mean value of $\Delta K_{th}$ was much higher than the experimental results when the residual shear stress was not considered. However, the mean value of $\Delta K_{th}$ had good agreement with the experimental data when the residual shear stress was taken into account. This phenomenon supported the previous analysis that the residual shear stress was not negligible and should be considered in analyzing the effect of residual stress on fatigue behavior. Therefore, it can be concluded that the effect of residual shear stress on the threshold stress intensity factor is significant and ignoring it can cause considerable deviation from the experimental results.

## 5. Conclusions

From the results obtained in this work, the following conclusions can be derived:

- The residual normal stresses $\sigma_{xx}$ and $\sigma_{yy}$ were tensile, while the residual shear stresses $\tau_{xy}$ were negative with the value changes in a range of −265~−211.5 MPa which demonstrated that the residual shear stress was not negligible.
- No clear and orderly tendency was found to conclude that the residual normal and shear stresses influenced the fatigue life. It was demonstrated that the residual stress in only one direction was not a suitable indicator for evaluating fatigue life. It is suggested that the fatigue life of a specimen was not directly dependent upon the surface shear and normal stresses.
- A linear relationship between the equivalent stress and the fatigue life was observed. The fatigue life increased by 39.4% as the equivalent stress decreased by 5.2% in this research. It was indicated

that the equivalent stress which considered residual shear and normal stress simultaneously was the dominate factor affecting fatigue life.

- The mean values of $\Delta K_{th}$ calculated by the proposed model had good accordance with the experimental data that were demonstrated by Mercer [29] and Clavel [30].
- The effect of residual shear stress on the threshold stress intensity factor was significant and ignoring it could cause a considerable deviation from the experimental results.

**Author Contributions:** Y.H. conceived and designed the study; Y.H. performed the experiments and analyzed the data; Y.H. wrote the paper; Z.L. reviewed this project and proposed constructive guidance to make the article more complete.

**Funding:** This work was support from the National Natural Science Foundation of China (No. 51425503 and 91860207). This work was also supported by grants from Taishan Scholar Foundation, the Major projects of National Science and Technology (No. 2019ZX04001031), the National Key Research and Development Program of China (2018YFB2002201), and Shandong Provincial Natural Science Foundation of China (ZR2019MEE073).

**Conflicts of Interest:** The authors declare no conflicts of interest.

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
