# Peer review of "Effects of Machining Induced Residual Shear and Normal Stresses on Fatigue Life and Stress Intensity Factor of Inconel 718"

_applsci, doi:10.3390/app9224750_

Round 1

Reviewer 1 Report

The paper concerns the problem of influence of residual shear and normal stresses on fatigue life and stress intensity factor.

The work is interesting and generally good presented. The Authors made necessary correction as suggested by the reviewer.

In my opinion, the paper is suitable for publication in this journal in presented form.

Author Response

Dear Editor and reviewers,

We appreciated the thorough reviews provided by the journal editor and the comments of reviewers on our manuscript titled " Effects of machining-induced residual shear and normal stresses on fatigue life and stress intensity factor of Inconel 718 " (ID: Applsci-632444). We thank the reviewers for the time and effort that they have put into reviewing the previous version of our manuscript. Their suggestions and comments enabled us to improve our work. Based on the instructions provided in your e-mail, we have uploaded the file of the revised manuscript. Appended to this letter is our point by point to the comments raised by the reviewers. We have clearly marked all improvements /corrections by using red font in the revised manuscript.

We would also like to thank you for allowing us to resubmit the revised copy of our manuscript.

Sincerely,

Yang Hua, Zhanqiang Liu*

Shandong University, China

Response to Reviewer #1 Comments

The paper concerns the problem of influence of residual shear and normal stresses on fatigue life and stress intensity factor.

The work is interesting and generally good presented. The Authors made necessary correction as suggested by the reviewer.

In my opinion, the paper is suitable for publication in this journal in presented form

Answer: Thank you very much for your comment.

Reviewer 2 Report

* English should be carefully checked, mainly for typos (e.g. line 66: literatureS, line 82: ARE corresponding, line 155: Since -> For, line 370: unit is missing, etc.) and for the choice of the tense.
* Line 207: f is the feed per revolution and not the feed rate.
* Isn't there an influence of polishing on the residual stresses as the depth of the residual stresses measurement is only 5 µm?
* Fig 8: I guess that "run" stands for "specimen", thus should be specified. No information on the cutting conditions of the corresponding specimens is provided.
* Table 5: There is only a selection of the results in the Table, why?
* Line 338 and Fig 10: I am not sure to agree with the analysis. I suspect a linear increase with Nf when looking at the plot. What are the correlation factors for the constant hypothesis you suggest and the linear increase hypothesis?
* Line 351 and Fig 11: R = 0.05 for Osinkolu and not R = 0.1, contrary to the legend and the indication on the figure; this should be specified. In addition, what is the point to compare results for different R values without taking that difference into account? The relevancy of the results of Osinkolu is not clear.

Author Response

Response to reviewer comments:

Nov 3rd, 2019

Dear Editor and reviewers,

We appreciated the thorough reviews provided by the journal editor and the comments of reviewers on our manuscript titled " Effects of machining-induced residual shear and normal stresses on fatigue life and stress intensity factor of Inconel 718 " (ID: Applsci-632444). We thank the reviewers for the time and effort that they have put into reviewing the previous version of our manuscript. Their suggestions and comments enabled us to improve our work. Based on the instructions provided in your e-mail, we have uploaded the file of the revised manuscript. Appended to this letter is our point by point to the comments raised by the reviewers. We have clearly marked all improvements /corrections by using red font in the revised manuscript.

We would also like to thank you for allowing us to resubmit the revised copy of our manuscript.

Sincerely,

Yang Hua, Zhanqiang Liu*

Shandong University, China

Response to Reviewer #2 Comments

Question 1: English should be carefully checked, mainly for typos (e.g. line 66: literatureS, line 82: ARE corresponding, line 155: Since -> For, line 370: unit is missing, etc.) and for the choice of the tense.

Answer: Thanks very much for your good suggestion first. We had corrected typos (e.g. line 66: literatureS, line 82: ARE corresponding, line 155: Since -> For, line 370: unit is missing, etc.) and carefully checked English in the revised manuscript. Furthermore, we thoroughly checked all the choice of tense in the whole manuscript. The changes had been marked with red font for easy tracking.

Question 2: Line 207: f is the feed per revolution and not the feed rate.

Answer: Thanks very much for your good suggestion first. We had changed the “feed rate” as “feed per revolution” in the revised manuscript.

Question 3: Isn't there an influence of polishing on the residual stresses as the depth of the residual stresses measurement is only 5 µm?

Answer: Thanks very much for your good question. As the penetration depth of X-rays for Inconel 718 was about 5 μm during measurements. It was not suitable to adopt the silicon carbide abrasive paper for polishing. In our current research, all the fatigue specimens were polished by using the polishing cloth to reduce the influence of surface roughness on fatigue performance. Consequently, there was less influence of polishing on the residual stresses.

Question 4: Fig 8: I guess that "run" stands for "specimen", thus should be specified. No information on the cutting conditions of the corresponding specimens is provided.

Answer: Thanks very much for your good suggestion first. Yes, “run” in Fig. 8 stood for “specimen”. To present a clearer explanation, the “run” in Fig. 8 was changed as “specimen” in our revised manuscript.

According to Moussaoui et al. (2015), the residual stresses were used as a basis to determine the milling plans of the procedure to conduct fatigue tests. To do so, the authors performed total six milling plans of procedure to observe the residual stresses influence on the fatigue life. Each plan of procedure was tested for two levels of loading: level 1 (σmax1) and level 2 (σmax2). For each level, two coupons were tested. In total, 24 fatigue tests were conducted. The cutting conditions of the corresponding specimens had been added in the Fig. I (Fig. 8 in the revised manuscript).

Reference

Moussaoui, K., Mousseigne, M., Senatore, J., Chieragatti, R. The effect of roughness and residual stresses on fatigue life time of an alloy of titanium. Int. J. Adv. Manuf. Tech. 2015, 78, 557–563.

Question 5: Table 5: There is only a selection of the results in the Table, why?

Answer: Thank you very much for your question first. The fatigue fracture morphologies of specimens under different machined surface conditions were observed by using SEM. It was found that all the specimens (the total 24 specimens) exhibited the similar fracture surface pattern. The crack occurred initially from the surface of specimens, then propagated into interior of machined surface. Therefore, we selected 12 specimens with clearer fracture surface pattern from the total 24 specimens to ensure the accuracy of calculation.

Question 6: Line 338 and Fig 11: I am not sure to agree with the analysis. I suspect a linear increase with Nf when looking at the plot. What are the correlation factors for the constant hypothesis you suggest and the linear increase hypothesis?

Answer: Thank you very much for your questions first. Fig. II (Fig. 11 in the revised manuscript) showed the relationship between the fatigue life Nf and SIF range ΔKth. From this figure, the values of ΔKth, calculated by the Moussaoui’s model, were in the range of 15.45~19.72 MPa  regardless of fatigue life. Whereas the values of ΔKth, calculated by the proposed model, varied in the range of 11.19~14.78 MPa  regardless of fatigue life, which were lower than the calculated results by Moussaoui’s model. As shown in the Fig. II (Fig. 11 in the revised manuscript), there was no obvious variation in the values of ΔKth with the increase of fatigue life for both the proposed model and Moussaoui’s model. Therefore, it was indicated that the ΔKth calculated by both Moussaoui model and the proposed model maintained a constant regardless of fatigue life.

The similar phenomenon was also observed by Liu et al. (2015), Yang et al. (2017), Li et al. (2017) and Sakai et al. (2002). They stated that ΔKth maintained a constant regardless of fatigue life.

Reference

Liu, X., Sun, C., Hong, Y. Effects of stress ratio on high-cycle and very-high-cycle fatigue behavior of a Ti–6Al–4V alloy. Materials Science and Engineering: A, 2015, 622, 228-235.

Yang, K., He, C., Huang, Q., Huang, Z. Y., Wang, C., Wang, Q., Zhong, B. Very high cycle fatigue behaviors of a turbine engine blade alloy at various stress ratios. International Journal of Fatigue, 2017, 99, 35-43.

Li, W., Zhao, H., Nehila, A., Zhang, Z., Sakai, T. (2017). Very high cycle fatigue of TC4 titanium alloy under variable stress ratio: Failure mechanism and life prediction. International Journal of Fatigue, 2017, 104, 342-354.

Sakai, T., Sato, Y., Oguma, N. Characteristic S–N properties of high‐carbon–chromium‐bearing steel under axial loading in long‐life fatigue. Fatigue Fracture of Engineering Materials Structures, 2002, 25(8‐9), 765-773.

Question 7: Line 351 and Fig 11: R = 0.05 for Osinkolu and not R = 0.1, contrary to the legend and the indication on the figure; this should be specified. In addition, what is the point to compare results for different R values without taking that difference into account? The relevancy of the results of Osinkolu is not clear.

Answer: Thank you very much for your suggestion first. We had amended the legend and the indication on Fig. III (Fig. 12 in the revised manuscript) according to your suggestion.

As shown in Fig. II (Fig. 12 in the revised manuscript), the value of ΔKth obtained by Osinkolu et al. (2003) was about 21.39 MPa  under the stress ratio R=0.05. In our current research, the value of ΔKth calculated by the proposed model was 13.45 MPa  under the stress ratio R=0.1, which was lower than the value of ΔKth=21.39 MPa  obtained under R=0.05 by Osinkolu et al. (2003). The similar phenomenon was observed by Sakai et al. (2002) and Liu et al. (2015) who demonstrated that the value of ΔKth decreased with the increase of stress ration. By comparing the results of the proposed model with the results of Osinkolu et al. (2003), we can validate the proposed model was correct.

Reference

Osinkolu, G.A., Onofrio, G., Marchionni, M. Fatigue crack growth in polycrystalline IN 718 superalloy. Mater. Sci. Eng. A 2003, 356, 425–433.

Sakai, T., Sato, Y., Oguma, N. Characteristic S–N properties of high‐carbon–chromium‐bearing steel under axial loading in long‐life fatigue. Fatigue Fracture of Engineering Materials Structures, 2002, 25(8‐9), 765-773.

Liu, X., Sun, C., Hong, Y. Effects of stress ratio on high-cycle and very-high-cycle fatigue behavior of a Ti–6Al–4V alloy. Materials Science and Engineering: A, 2015, 622, 228-235.

This manuscript is a resubmission of an earlier submission. The following is a list of the peer review reports and author responses from that submission.

Round 1

Reviewer 1 Report

The paper concerns the problem of influence of residual shear and normal stresses on fatigue life and stress intensity factor.

The work is interesting and generally good presented, but a few issues must be corrected for greater readability and usability.

The main remarks to presented work:

The review of the literature connected to the subject of the article was made at the lowest acceptable level. 1a in combination with Fig. 1b and 1c are difficult to interpret. What stress state is represented by the cube marked in figure 1a (1b or 1c)? It should be presented in other way. There is no consistency in the variable markings used in the work. For example: σx - σxx and so on. There are definitions of angles φ and ψ in the paper. There is no definition of angle α. Angle markings in brackets (for example: s(φ=0)) used by the Authors make reading difficult. Using subscripts or superscripts would be clearer for the Readers. Please explain values from Table 3. What is correct: V or Vc? What is presented on figure 5? Mean value? There were 6 samples for each machining condition. Shear stress cannot be “compressive”! (For example: Line 237 and further in the text). There are no results of residual shear and normal stress measuring for all samples. Are they the same for all samples? Results and their discussion presented in Point 4.2 are incorrect. Results presented on Fig. 6 are under 30 thousand and over 60 thousand of cycles. What was mentioned in the discussion on this figure? Mean value? And the most important: an identical residual stress state was measured in each sample? Please explain what were presented on figure 8. Where did the data come from? What are “Runs”? Where the four results for each “Run” come from? What is the “Model” and what equation describing it? The presentation of table 5 is insufficient. Why only 12 from 24 samples are presented? And again: the same residual stress values for each sample? Why Vc=50 m/min has been highlighted? The presented material is not satisfactory in terms of own fatigue tests.

Other remarks to presented work:

There is mistake in value of φ (Line 122). There is mistake in Line 189 (“worpiece”).

Author Response

Dear Editor and reviewers,

We highly appreciated the thorough reviews provided by the journal editor and the comments of reviewers on our manuscript titled "Machining-induced residual shear and normal stresses on fatigue life and stress intensity factor of Inconel 718" ( Manuscript ID: applsci-593671). We thank the reviewers for their time and effort that they have put into reviewing the previous version of our manuscript. Their suggestions have enabled us to improve our work. Based on the instructions provided in your e-mail, we have uploaded the file of the revised manuscript. Appended to this letter is our point by point to the comments raised by the reviewers. We have clearly marked all improvements /corrections by using red font in the revised manuscript.

We would also like to thank you for allowing us to resubmit the revised copy of our manuscript.

Sincerely,

Yang Hua, Zhanqiang Liu*

Shandong University, China

Author Response

(The authors gave the same response as above.)

Author Response

(The authors gave the same response as above.)

Round 2

Reviewer 2 Report

The main comments from the previous revisistill apply. Authors skipped comment 3 on the previous revision, which was a major concern, in their rebuttal.

Major issues in the design of experiments, which are not addressed.

Reviewer 3 Report

Unfortunately, the authors could not answer any of the concerns I have about the work. T